# The Spatial Concentration and Dispersion of Homicide during a Period of Homicide Increase in Brazil

Spencer P. Chainey [1,*] and Franklin Epiphanio Gomes de Almeida [2]

1   UCL Jill Dando Institute of Security and Crime Science, University College London, 35 Tavistock Square, London WC1H 9EZ, UK
2   Mato Grosso Military Police, Avenue Historiador Rubens de Mendonça, 6135, Cuiaba 78049-090, MG, Brazil; franklin.almeida.18@alumni.ucl.ac.uk
*   Correspondence: s.chainey@ucl.ac.uk

**Abstract:** This study applies the principles of measuring micro-place crime concentration and the spatial dispersion of crime increase to the geographic unit of cities in Brazil. We identify that a small number of cities account for a large cumulative proportion of homicides, and that during a period of homicide increase 30 cities out of 5570 accounted for the equivalent national increase in homicides. The majority of the 30 cities were not established high homicide cities but instead were new emerging centers of homicide that neighbor high homicide cities. We suggest the findings can be used to better target effective programs for decreasing homicides.

**Keywords:** spatial crime concentration; homicide; Brazil; Latin America; neighboring cities

## 1. Introduction

Brazil consistently accounts for one in seven of all recorded homicides in the world, yet represents only 3% of the world's population [1,2]. Over recent decades, homicide levels have increased in Brazil, rising from 48,219 homicides in 2007 to 65,602 in 2017 [3]—a rate of 31.6 homicides per 100,000, compared to a global rate of 6.1 [3,4]. In the wider context of Latin America, Brazil's high levels of homicide are representative of the issues of violence in the region, and because of the significant role the country plays in the region's social development and prosperity, research on homicide in Brazil is valuable for understanding Latin America's homicide issues [5].

National statistics on homicide can, however, mask important geographic variations within a country. For example, in recent years some cities in Brazil such as Fortaleza and Rio Branco have followed the national trend and have experienced increases in homicide. However, several other cities such as São Paulo and Belo Horizonte have not experienced increases and in some cases the level of homicide has decreased. Reasons for the variation in homicide levels in Brazil have been associated with differences in social disorganization, opportunities for drug trafficking, exploitation of natural resources, and conflicts between organized crime groups [5–15]. In this paper we build on the research about the variation in homicides in Brazil by analyzing the geographic dynamics of homicide trends across the country, and by doing so illustrate how the methods we use can be applied to examining crime patterns across a country. The methods we use involve an examination of the spatial concentration of homicides in Brazil at the city level and an examination of the spatial dispersion of homicides across these cities during a period of homicide increase.

A significant body of research shows that crime spatially varies, and concentrates in a relatively small number of places. This observation is so consistent that Weisburd [16] proposed a *law of crime concentration*, suggesting that about 1% of places account for a cumulative proportion of 25% of crime, and that about 4% of places account for a cumulative proportion of 50% of crime. Weisburd's law for examining the spatial concentration of crime has a focus towards micro-places, such as street segments within a city. Studies

examining micro-place patterns of spatial concentration have then informed the design and targeting of successful interventions that counter these patterns [17,18]. We propose that the process for measuring the spatial concentration of crime can also be applied to the larger geographic unit of cities to help facilitate the empirical examination of crime across a country. Specifically, with regards to the current study, the analysis of the spatial concentration of homicides in Brazil can identify whether only a few cities account for a large cumulative proportion of homicides across Brazil. An examination of the spatial concentration of homicides for cities across a country may in turn better inform the targeted implementation of national or state government homicide reduction strategies.

A consistent finding from several other studies that have examined the geography of crime is that during periods of crime increase or decrease, a small proportion of places are responsible for these changes in crime, with the places that previously registered the highest levels of crime accounting for the largest changes [19–24]. These studies have only focused on examining micro-place or neighborhood patterns of crime. We propose the techniques used in these studies for examining areas responsible for changes in crime can also be applied to the meso-scale of cities to determine if a national increase in crime is associated with only a small number of cities. Additionally, these techniques can determine if it is those cities that previously recorded the highest levels of crime were the cities where most increases in crime were experienced, or if crime has dispersed to other areas. By doing so, we present the first ever findings that have applied the combination of these techniques of spatial concentration and the dispersion of crime to meso geographic units (Saraiva et al.[25] have examined the spatial patterning of homicides across Brazil over a 10-year period using cities as the geographic unit but did not use the statistical measures of spatial dispersion for calculating and identifying those cities that contributed most to the increases in homicide in Brazil). By applying these methods to homicide, the study reveals findings on patterns of homicide across a country during a period of increase, and more specifically contribute to our understanding of homicide patterns in Brazil.

In the section that follows we discuss homicide trends in Brazil from which we highlight the potential value in examining the geographic dynamics of these trends within the country. In section three we describe the methods used for examining micro-place spatial concentration and the dispersion of crime increase, before describing the data used and how these methods were applied in the current study. Results are presented in section five, followed by a discussion of the findings, limitations of the study, and conclusions.

## 2. Homicide in Brazil

Brazil's high levels of violence are a continually debated topic that shape everyday social interactions in the country [26–29]. Although violence is manifested in different ways, homicides are the maximum expression of the problem of violence in Brazil. In 2017, a new record of 65,602 homicides for a single year was registered in Brazil, representing more than 179 homicides per day [3]. This placed Brazil as the deadliest country in the world in absolute numbers, with a homicide rate that was 30 times higher than Europe's [30]. Homicides in Brazil are mostly committed by and against young black men living in impoverished conditions [31,32]. High homicide levels also restrict the country's prosperity, from youths killed being lost to the labor market, to its effect on the prices of goods and services [12]. Homicide in Brazil has an annual social cost that is equivalent to 5.9% of the country's Gross Domestic Product, which in 2017 represented USD 110.9 billion [1,12].

Most studies that have examined geographic variations in homicide in Brazil (and Latin American countries more widely) have focused on examining the influence of structural conditions such as social inequality and poverty on these homicide patterns [33–36]. More recently, other studies have additionally highlighted weaknesses in institutional legitimacy, government ineffectiveness, the rule of law, impunity, and development as factors that influence homicide levels [37–45]. Violence in Brazil is also considered to be associated with the drugs trade, with Brazil playing an important role in the trafficking of drugs from neighboring countries (such as Colombia) to European, African, Asian,

and Australasian markets [46,47]. Coinciding with the role of Brazil as a hub for drug trafficking has been the geo-economic expansion of the largest criminal organizations in the country—the Primeiro Comando da Capital (PCC) and the Comando Vermelho (CV). These groups, previously most present in the southeast and south regions of Brazil, have expanded their activities to the north and northeast of the country with allies based in these regions [12,48]. Collectively, these studies have provided valuable insights into the variation and dynamics of homicide in Brazil. In the current study we specifically examine the geographic dynamics of homicide in Brazil with the aim of generating findings that add to this previous research.

Homicides in Brazil are heavily concentrated in urban areas [49]. In Brazil, there are 5570 cities [50]. Previous research using a data sample of 4491 cities found that 16% of Brazilian cities accounted for 73% of all homicides between 1991 and 2010, variations in changes in homicide ranged between increases of 207% and decreases of 26%, with differences in the social disorganization of cities (such as lack of effective formal social control) being the reason for the variations in homicide levels [13]. More specifically, homicides among men aged 20–39 years have been observed to be greatest in highly urbanized municipalities [49], and tend to be lower where Brazil's cash transfer (poverty alleviation) program was present [29,51]. The level of homicides also tends to be lower where stronger restrictions on access to and use of firearms is imposed [29,52].

Other researchers have highlighted how certain changing dynamics within Brazil affect homicide levels in the country. This includes changes in economic conditions and urban development that has resulted in cities near international borders and coastal cities experiencing increases in homicide because of changes in their functional status [6]. To help provide a basis for supporting research that examines homicide patterns in Brazil, Waiselfisz [14,15] proposed five categories to define Brazilian cities where increases in homicide have been observed: new poles of growth, border cities, new frontiers, seaside cities, and cities in the Marijuana Polygon. Waiselfisz [14,15] identified that new poles of growth tended to be smaller or medium-sized cities where a process of economic decentralization and development from the late 1990s resulted in investment in this type of city, attracting many people, and with it more opportunities for crime. Border cities are those located near to international borders, and served as gateways for smuggling, and the trafficking of weapons, drugs, and people. New frontiers are cities located in predominantly rural regions, characterized by issues associated with illegal logging and mining, conflicts over land tenure, and the exploitation of local communities often because of large agricultural and national development ventures that demand 'unoccupied' land. Seaside cities are those that have port facilities, making them a hub for outbound illicit drugs and inbound firearms, in addition to attracting tourists who could be vulnerable to crime. The Marijuana Polygon includes cities across a region located at the junction of Bahia, Pernambuco, Alagoas, Sergipe, and Ceará states, responsible for cannabis production in Brazil. Ceccato and Ceccato [8] have highlighted that small cities in Brazil have experienced the largest increases in homicide, with Steeves et al. [53] suggesting these increases in violence are associated with economic prosperity in smaller cities. In contrast, several large metropolises in Brazil have experienced decreases in homicide, such as São Paulo where new public safety programs and improvements in policing methods have coincided with decreases in homicide [54,55]. Ingram and Da Costa [10] have also observed that if an area experiences a high level of homicide this can result in an increase in homicides in nearby areas.

This synthesis of the research on homicide patterns in Brazil, its spatial distribution and trends, highlights the multi-faceted nature to homicide in Brazil. Homicides in Brazil are not equally distributed across the country, are most present in cities rather than rural areas, and are subject to changing dynamics in the country. Countering the problem of homicide not only requires federal action, but also effective action at state and city levels of government. Prioritizing resources to areas of most need, and tailored program delivery to addressing the conditions that reduce homicides in different settings are considered to be key factors in the delivery of effective homicide reduction programs [56]. To ensure

homicide programs in Brazil are effectively implemented requires an appreciation of how homicide patterns spatially vary across the country, especially during periods when homicide levels increase. Methodological developments in examining the geographic concentration of crime and the dispersion of crime during periods of crime increase can offer a means for adding to existing understanding of patterns of homicide in Brazil.

### 3. Spatial Concentration and Emerging Problem Areas

The study of the geography of crime has increasingly focused on the micro-place (such as the street segment) as the geographic unit of analysis. A consistent finding from numerous studies on micro-place patterns of crime is that a small number of places are responsible for a large proportion of crime [16,57–59]. Many of these studies have applied the bandwidths suggested by Weisburd [16] for comparing between settings and different crime types. These bandwidths are used to calculate the proportion of places that are responsible for a cumulative proportion of 25% of crime and for a cumulative proportion of 50% of crime. The results from micro-place analysis of crime have then been used to determine where to target interventions for decreasing crime, such as hot spot policing and problem-oriented policing programs [17,18]. Use of the same methodological process for examining micro-place patterns of crime may help to better formalize the examination of the spatial concentration of homicide for larger geographic units, such as cities within a country.

When crime increases, attention focuses to identifying the areas where increases have been greatest. Micro-place studies of crime show that when crime increases, the largest increases take place in the areas where crime levels were already high [21]. To help measure and identify the areas that are most responsible for an overall increase in crime, Ratcliffe [23] developed a series of dispersion indices to indicate if an overall crime increase is associated with only a small number of areas or if the increase is a spreading (emergent) problem. Ratcliffe's dispersion measures do not, however, determine if the crime increase is associated with an increase in areas where most crime previously occurred. To address this, Chainey and Monteiro [21] developed the Crime Concentration Dispersion Index (CCDI) to determine whether, during a period of crime increase, areas of high crime concentration were responsible for the increase or if other areas were responsible. To date, these measures for crime dispersion have only been applied to micro-place and neighborhood geographic units (because of recent research focus to micro-place analysis of crime), yet their application is suitable for any size of geographic unit, such as cities. The techniques developed by Ratcliffe [23] and Chainey and Monteiro [21], therefore, can be used to determine whether an increase in crime in a country was mainly associated with only a small number of cities being responsible for the increase, and whether cities with the highest levels of crime were mainly responsible for a national increase in crime.

In this study, we use techniques that have been applied to examine crime at the micro-place level to the patterns of homicide at the meso-place level (i.e., cities across a country). We recognize that city size will influence the overall homicide level in a city but similar to analysis of crime at micro-place levels, micro-place geographic units (e.g., street segments) substantially vary in size and have provided valuable insights into geographic patterns of crime. In micro-place analysis of crime, we would anticipate that longer street segments would account for a larger number of crimes than shorter street segments. Micro-place analysis of spatial concentration does not normalize crime count data to rates based on geographic unit size, and yet has provided valuable insights about the spatial patterning of crime. We anticipate that the larger cities in Brazil will account for the larger number of homicides, but also anticipate the analysis will provide valuable insights into geographic patterns of homicide across Brazil. Cities in Brazil are based on similar institutional frameworks and share relatively similar cultural heritages [10], which in turn make them suitable units for comparative analysis.

The research was guided by testing three hypotheses: Homicide is highly concentrated across cities in Brazil; a small number of cities in Brazil were responsible for recent national

increases in homicide; the homicide increase in Brazil was associated with cities that previously recorded the highest levels of homicide. We anticipate that the results that are generated from testing these hypotheses provide new insights into the geographic dynamics of homicide in Brazil. To posit these results in the context of previous research that has examined the patterning of homicides in Brazil, we aim to review our results in the discussion section against Waiselfisz's [14] five categories for defining cities in Brazil where increases in homicide have been observed.

### 4. Data and Methods

The unit of analysis was cities in Brazil, of which there were 5570 cities. The number of homicides in each city between 2007 and 2017 was extracted from the Mortality Information System of the Brazilian Ministry of Health [60]. These data refer to the occurrence of violent deaths, including intentional incidents, robberies that resulted in a homicide, and police killings. The data do not indicate the motivation for the homicide, such as if the intentional homicide was associated with the drugs trade and violence between criminal groups. Population data for each city were extracted from the Brazilian Institute of Geography and Statistics for the year 2017 [50]. Cities were organized into five size groupings to determine if patterns of homicide were particular to city size, following the method used by Duarte et al. [49]: Small cities I ($\leq$20,000 inhabitants; n = 3802); small cities II (from 20,001 to 50,000; n = 1103); medium cities (from 50,001 to 100,000; n = 355); big cities (from 100,001 to 900,000; n = 293), and metropolises ($\geq$900,001; n = 17). The number of cities that accounted for 25% and 50% of Brazil's homicides in each year between 2007 and 2017 were calculated. This process was performed in Microsoft Excel (following the procedure described by Chainey [61]) and involved arranging the data on homicides in each city for each year into a table and then rank ordering the data for each year from the highest to the lowest number of homicides in each city. Then, the percentage of homicides in each city relative to all homicides was calculated, followed by the calculation of the cumulative percentage for each city, across all the cities. This procedure is the same procedure that is used for examining micro-place crime concentration albeit where the geographic unit of analysis is the street segment. We calculated the level of homicide spatial concentration for each year to determine if this measure changed over time and for a period when homicides in Brazil had increased. We refer hereafter to cities that accounted for 25% of homicides as high homicide cities—HHCs.

The study's examination of the dispersion of homicide was focused towards examining the change in crime between the two years for the most recent period that data were available (i.e., 2016 to 2017) for purposes of being up to date, and because 2017 was when there was a peak in the number of homicides observed in Brazil. The spatial dispersion of homicide increase in Brazil was analyzed using Ratcliffe's Dispersion Calculator [23]. The Dispersion Calculator compares the changes in crime between two time periods (t1 and t2, i.e., 2016 and 2017) for each geographic unit in the study area. When a crime increase in a study area has been observed, the Dispersion Calculator determines whether the increase in the study area was related to only a small number of places experiencing a large increase in crime, or whether the increase was associated with smaller increases across a large number of places within the study area. The Dispersion Calculator works by ordering the geographic units in the study area by the level of crime increase in each unit, and then removes these ordered units one at a time (starting with the unit with the highest crime increase) and removes their respective incidence of crime from the total for the study area. For each iteration, the increase in crime across the study area (based on all remaining geographic units) is recalculated. By doing so, the Dispersion Calculator determines the point at which the removal of the geographic units that experienced the highest increases in crime generates a revised study area change in crime measure that shows no increase in crime across the remaining geographic units between the two time periods. Some other geographic units may have experienced an increase in crime, but these increases would

have been smaller and could have been offset by the places where decreases in crime were observed.

The Dispersion Calculator generates two indices: The Offense Dispersion Index (ODI) and the Non-Contributory Dispersion Index (NCDI). The ODI is the proportion of geographic units that must be removed from the study-wide calculation before the increase in crime across the study area is transformed to a decrease, or at least a no-change steady state is observed (for more details see Ratcliffe [23]). The ODI is calculated using the following equation:

ODI = (n of geographic units that must be removed from the study-wide calculation before the increase across the study area is transformed to a decrease)/(n of all geographic units in the study area)

The ODI (ranging from zero to one) determines the smallest proportion of geographic units that alone account for a study area equivalent increase in crime. For example, in a study area that experienced a 20% increase in crime and consisted of 100 geographic units, the ODI is a measure of the smallest proportion of geographic units that alone accounted for the study area's 20% increase in crime. For instance, if five geographic units experienced large increases in crime, and the total increase in these five geographic units was equivalent to the study area's 20% increase in crime, the ODI would be 0.05 (i.e., 5/100). If ten geographic units experienced the largest increases that in total was equivalent to the study area's 20% increase in crime, the ODI would be 0.1 (i.e., 10/100). An ODI value close to zero indicates that only a small number of geographic units experienced an increase in crime that was equivalent to the study area's increase.

The NCDI (ranging from zero to one) indicates the proportion of other geographic units that are a concern and is a measure of the proportion of the other geographic units in the study area that experienced an increase in crime. The NDI is calculated using the following equation:

NCDI = (n of geographic units in the study area that experienced an increase in crime, but not including those units included in the ODI calculation that experienced the highest increases in crime)/(n of all geographic units in the study area)

For example, following on from the example of a study area that consists of 100 geographic units, if 35 other geographic units experienced increases in the crime (i.e., in addition to the five geographic units that experienced the greatest increases in crime), the NCDI for this study area would be 0.35 (i.e., 35/100). An NCDI value close to zero indicates the crime increase has not been observed in many other geographic units. To date, although the ODI and NCDI can be applied to any size of geographic unit and any size of study area it has only been applied to the geographic units of street segments and neighborhoods (for examples see Chainey and Monteiro [21] and Ratcliffe [23]). In the current study we apply these measures to the geographic unit of cities. ODI and NCDI values for homicide were calculated for the whole country of Brazil, and for each city group categorization. The Dispersion Calculator was also used to identify the specific cities that accounted for the ODI value (i.e., the cities that experienced the highest levels of homicide and that collectively accounted for an increase in homicide that was equivalent to the increase in homicide experienced in Brazil between 2016 and 2017). We refer to these cities hereafter as emerging problematic cities—EPCs.

The Crime Concentration Dispersion Index was used to determine if cities where homicide levels were previously high were the cities responsible for the crime increase (following the methodology described by Chainey and Monteiro [21]). The CCDI is the ratio of the homicide increase in EPCs (calculated using the Dispersion Calculator) that were *not* identified as cities with high homicide levels (i.e., non-HHC EPCs), and the homicide increase in the high homicide cities (i.e., HHCs). For the non-HHC EPCs, the total increase in homicides experienced between 2016 (t1) and 2017 (t2) was calculated, and then averaged per non-HHC EPC. Similarly, for the high homicide cities, the total increase in

homicides experienced between t1 and t2 for all HHCs was calculated, and then averaged per HHC. The CCDI is calculated using the following equation:

CCDI = (Crime increase between t1 and t2 per non-HHC EPC)/(Crime increase between t1 and t2 per HHC)

A CCDI value of less than one indicates that high homicide cities contributed more to the increase than other emerging problematic cities (i.e., the non-HHC EPCs). The closer the CCDI is to zero, the less the need for targeting resources to cities other than HHCs. A CCDI of one indicates that HHCs and other non-HHC EPCs equally contributed to the increase in homicides, meaning that HHCs and these new emerging problematic cities require attention if homicide levels are to be decreased. A CCDI of greater than one indicates new emerging problematic cities (i.e., non-HHC EPCs) contributed more to the increase than high homicide cities.

To assist in the presentation of the results we also organized the results for cities by the regional geography of Brazil (north, northeast, southeast, south and midwest). We used these regions in order to be consistent with the definition of regions of Brazil that are most commonly used, particularly in studies of homicide in Brazil [12].

## 5. Results

Homicides were highly spatially concentrated across cities in Brazil. When considering the total number of homicides that were observed between 2007 and 2017, only 15 cities (equivalent to 0.27% of all cities in Brazil) accounted for 25% of all homicides, and 95 cities (equivalent to 1.7% of all cities) accounted for 50% of all homicides. The level of spatial concentration for homicides across cities in Brazil changed very little between 2007 and 2017, increasing from 0.20% to 0.34% of cities accounting for 25% of all homicides (Table 1), even though homicides increased by 36.9% over this period. All the cities that accounted for 25% of homicides between 2007 and 2017, and for each year within this period, were either metropolises or big cities.

**Table 1.** The concentration of homicides in Brazil between 2007 and 2017, and the types of cities that were HHCs.

| Year | Number (and Proportion) of Cities Accounting for 25 Percent of All Homicides | High Homicide Cities (HHCs) | |
| --- | --- | --- | --- |
| | | Metropolis | Big Cities |
| 2007 to 2017 | 15 (0.27) | 14 | 1 |
| 2007 | 11 (0.20) | 11 | 0 |
| 2008 | 12 (0.22) | 12 | 0 |
| 2009 | 13 (0.23) | 13 | 0 |
| 2010 | 13 (0.23) | 11 | 2 |
| 2011 | 14 (0.25) | 13 | 1 |
| 2012 | 14 (0.24) | 14 | 0 |
| 2013 | 15 (0.27) | 14 | 1 |
| 2014 | 16 (0.29) | 14 | 2 |
| 2015 | 17 (0.31) | 14 | 3 |
| 2016 | 20 (0.36) | 15 | 5 |
| 2017 | 19 (0.34) | 15 | 4 |

Figure 1 shows that cities in the northeast of Brazil increased in their contribution as HHCs between 2007 and 2017. HHCs in the northeast region included Fortaleza, Salvador and Recife. More cities in the southeast region became HHCs in 2016 and 2017, with these

additional cities including São Gonçalo and Duque de Caxias located in the state of Rio de Janeiro.

**Figure 1.** HHCs in Brazil, by region.

Between 2007 and 2017, each year experienced an increase in homicides with the exception of 2011 and 2015 (Table 2). The largest proportionate increases in homicides were experienced in small and medium-sized cities. Metropolises, although accountable for the largest proportion of homicides in Brazil, experienced homicide increases of only 5.3%. Small cities (category I), small cities (category II), and medium cities experienced homicide increases of 83.8%, 91.8%, and 79.7% respectively. Between 2016 and 2017, small cities (categories I and II) experienced the greatest proportionate increases in homicide.

**Table 2.** Homicide and the change in homicide by city category in Brazil from 2007 to 2017.

| Year | Brazil (n = 5570) | Small Cities I (n = 3802) | Small Cities II (n = 1103) | Medium Cities (n = 355) | Big Cities (n = 293) | Metropolis (n = 17) |
|---|---|---|---|---|---|---|
| 2007 | 47,228 | 3877 | 5394 | 4602 | 19,410 | 13,945 |
| 2008 | 49,545 | 4122 | 5801 | 5043 | 20,382 | 14,197 |
| 2009 | 50,927 | 4408 | 6435 | 5419 | 20,567 | 14,098 |
| 2010 | 51,938 | 4381 | 6603 | 5539 | 21,015 | 14,400 |
| 2011 | 51,785 | 4453 | 6884 | 5679 | 20,989 | 13,780 |
| 2012 | 55,890 | 4944 | 7314 | 6126 | 22,711 | 14,795 |
| 2013 | 56,308 | 5015 | 7725 | 6462 | 22,506 | 14,600 |
| 2014 | 59,391 | 5514 | 8305 | 6983 | 23,510 | 15,079 |
| 2015 | 58,278 | 5808 | 8719 | 7105 | 22,425 | 14,221 |
| 2016 | 61,531 | 6628 | 9534 | 7798 | 23,741 | 13,830 |
| 2017 | 64,660 | 7125 | 10,346 | 8270 | 24,228 | 14,691 |
| Change 2007–2017 (%) | 36.9 | 83.8 | 91.8 | 79.7 | 24.8 | 5.3 |
| Change 2016–2017 (%) | 5.1 | 7.5 | 8.5 | 6.1 | 2.1 | 6.2 |

Table 3 shows the ODI and NCDI values for Brazil. The homicide ODI for this period was 0.005 indicating that a small number of cities (n = 30) accounted for the equivalent national increase in homicides of 5.1% in Brazil between 2016 and 2017. That is, 30 of the 5570 cities in Brazil were identified has experiencing the highest increases in homicide between 2016 and 2017. In these 30 cities, homicides increased from 9975 in 2016 to 13,106 in 2017. Between the same two years, homicides increased in Brazil from 61,531 to 64,660, a 3129 numeric increase and 5.1% increase. The increase of 3131 homicides in the 30 cities that experienced the highest increases was equivalent to the Brazil-wide increase in homicides experienced between 2016 and 2017. The homicide NCDI for this period was 0.390 and indicated that many other cities (n = 2172) experienced increases in homicide between 2016 and 2017. The ODI and NCDI calculations were repeated for cities within each city size category, with very little difference being observed in the patterns for each category compared to that observed nationally—a small number of cities (in each city size category) accounted for the equivalent increase in homicides between 2016 and 2017, but a large number of other cities (in each city size category) had also experienced increases in homicide. For example, for the city size category of small cities (category II) only 67 of the 1103 cities accounted for the equivalent increase in homicides of 8.5% between 2016 and 2017 in this category (ODI = 0.061), but almost half of the cities in this category (n = 496; NCDI = 0.450) also experienced increases in homicide.

**Table 3.** ODI and NCDI values for homicide between 2016 and 2017 for Brazil and for city size categories.

|  | Brazil | Small Cities I | Small Cities II | Medium Cities | Big Cities | Metropolis |
|---|---|---|---|---|---|---|
| ODI | 0.005 | 0.017 | 0.061 | 0.037 | 0.017 | 0.056 |
| NCDI | 0.390 | 0.331 | 0.450 | 0.442 | 0.447 | 0.412 |

The CCDI value for Brazil was 1.129, indicating the national increase in homicides was more associated with cities that were not HHCs, rather than HHCs being responsible for the increase. That is, cities other than the cities where homicide levels had previously been the highest emerged as problematic areas for homicide. The contribution of emerging problematic cites, other than HHCs to the increase in homicide in Brazil between 2016 and 2017 is further illustrated in Table 4. Between 2016 and 2017, homicides in the high homicide cities in Brazil (n = 19) increased by 6.8%. Other cities that were mainly accountable for the increase in homicides in Brazil between 2016 and 2017 (i.e., non-HCC EPCs) (n = 22) experienced a homicide increase of 56.0%.

**Table 4.** Increase in homicides in high homicide cities in Brazil and the increase in homicides in emerging problematic cities that were not HHCs.

|  | Homicides 2016 | Homicides 2017 | Change 2016 to 2017 (%) |
|---|---|---|---|
| All HHCs | 15,187 | 16,219 | 1032 (6.8%) |
| Non-HHCs identified as EPCs | 2411 | 3760 | 1349 (56.0%) |

Table 5 lists the 30 cities in Brazil, by city size category, that accounted for the equivalent national increase in homicides of 5.1% between 2016 and 2017. Each city is listed by its percentage increase in homicide (and the region in which it is located). Most cities that contributed to the national increase were big cities rather than metropolises. The metropolises listed were all HHCs with the exception of Campinas. Several cities experienced increases in homicide of over 100%, including Barreiras, Gravatá, Gravataí, and Horizonte. The medium-sized cities that were mainly accountable for the equivalent national increase in homicides were all located in the northeast of Brazil. HHCs that were not included in this group of 30 cities (listed with their percentage change in homicide between 2016 and

2017) were Belém (−2.6%), São Paulo (−17.2%), Porto Alegre (−16.0%), Brasília (−19.7%), Goiânia (−3.5%), Natal (4.5%), Belo Horizonte (−12.8%), São Luís (−15.8%), Ananindeua (5.6%), Curitiba (−19.6%), and Nova Iguaçu (7.6%). Eight of these 11 HHCs experienced decreases in homicide in the year when Brazil recorded its highest levels of homicide.

**Table 5.** Cities in Brazil that accounted for the equivalent national increase in homicides between 2016 and 2017.

| | Cities (Percentage Increase in Homicides; Region) | |
|---|---|---|
| Metropolises | Campinas (27.3%; southeast) | *Recife* (24.4%; northeast) |
| | *Fortaleza* (84.3%; northeast) | *Rio de Janeiro* (10.0%; southeast) |
| | *Maceió* (9.2%; northeast) | *Salvador* (3.9%; northeast) |
| | *Manaus* (18.5%; north) | *São Gonçalo* (9.1%; southeast) |
| Big cities | Altamira (52.0%; north) | Gravataí (115.8%; south) |
| | Alvorada (58.5%; south) | Ilhéus (53.7%; northeast) |
| | Barreiras (221.1%; northeast) | Maracanaú (61.3%; northeast) |
| | Cabo de Santo Agostinho (31.4%; northeast) | Paulista (52.1%; northeast) |
| | Cariacica (32.7%; southeast) | Rio Branco (37.3%; north) |
| | Caucaia (83.7%; northeast) | Serra (16.8%; southeast) |
| | *Duque de Caxias* (34.5%; southeast) | Sobral (76.5%; northeast) |
| | Florianópolis (73.2%; south) | Vitória (72.6%; southeast) |
| Medium-sized cities | Abreu e Lima (97.4%; northeast) | Horizonte (120.0%; northeast) |
| | Ceará-Mirim (49.4%; northeast) | Ipojuca (90.0%; northeast) |
| | Gravatá (111.1%; northeast) | Pacajus (90.9%; northeast) |
| Small cities II | - | |
| Small cities I | - | |

Cities in italics were high homicide cities.

Figure 2 shows the distribution of the 30 cities that accounted for the equivalent national increase in homicides of 5.1%. Over half of these 30 cities were located in the northeast region of Brazil. Almost all the cities were located along the Atlantic coast, however, it is noted that the majority of urban settlements in Brazil are located along or near to the Atlantic coast, reflecting the lack of properly developed transport routes that extend into the interior from what is referred to as the Grand Escarpment that dominates much of Brazil's coast [62].

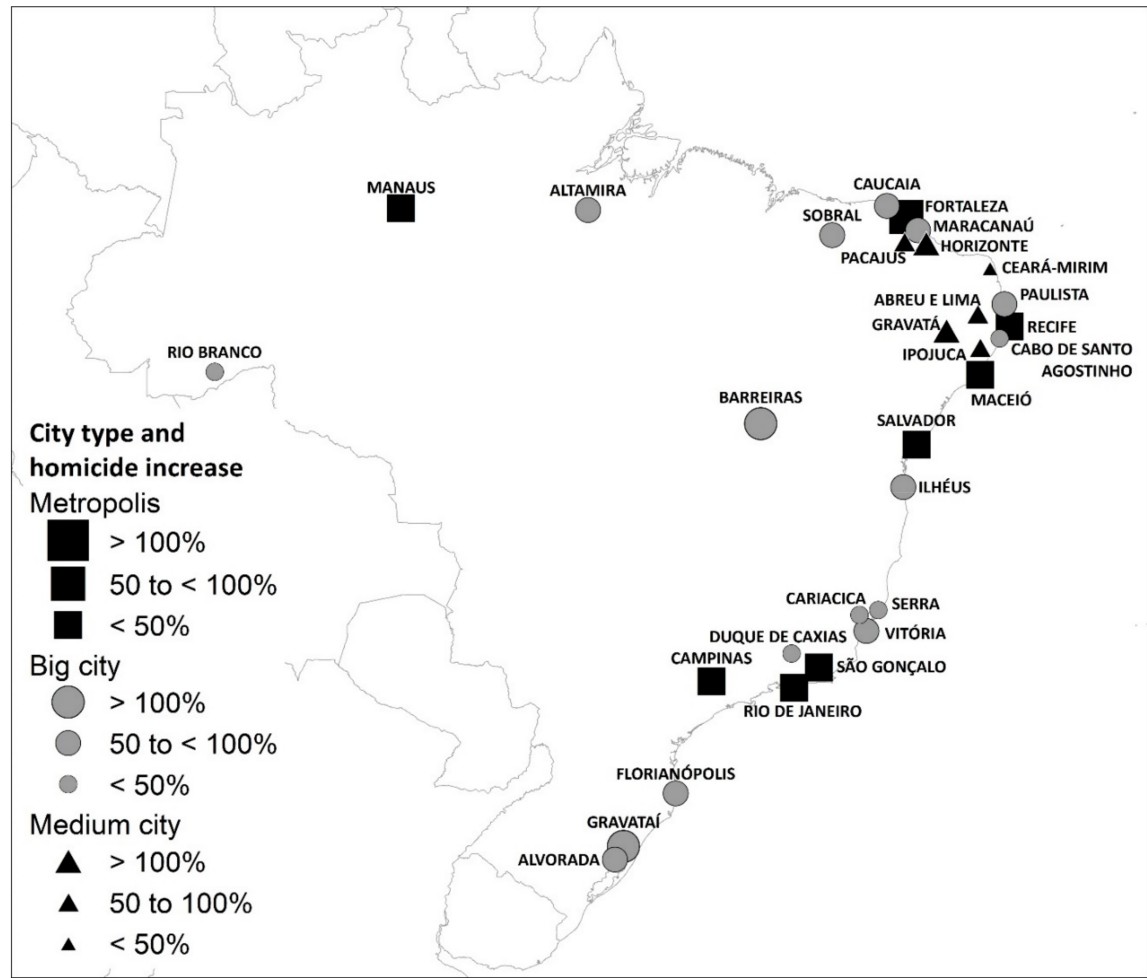

**Figure 2.** Cities in Brazil (and their respective homicide increase) that accounted for the equivalent national increase in homicides between 2016 and 2017.

## 6. Discussion

Brazil experiences some of the highest homicide levels in the world. In the current study the analysis was guided by testing three hypotheses, with the first of these stating that homicide is highly concentrated across cities in Brazil. This proved to be the case with no more than 20 of Brazil's 5570 cities (0.36%) being responsible for at least a quarter of all homicides in any year. This level of spatial concentration is comparable to the patterns of homicide concentration observed at micro-places (i.e., street segments) within cities across Latin America [58] and suggests a consistency in the spatial concentration of crime for geographic units across geographic scales. Similar to street segments, cities vary in size. It was apparent that city size was a factor in determining cities in Brazil that experienced the most homicides—for each year from 2007 to 2017, at least three-quarters of the high homicide cities in Brazil were metropolises. Studies examining micro-place concentrations of crime do not examine whether street segment length is a determining factor in identifying streets that experience the highest concentrations of crime. We applied this measurement principle for examining the spatial concentration of homicide across Brazil using cities as the unit of study but recommend further research on the spatial concentration of crime across scales that normalize for the size of the geographic unit. This could include the examination of crime rates (e.g., crimes per kilometer of street, crimes per 1000 city population) to determine those geographic units that contribute to the highest quartile of crime rates.

In 2017, Brazil experienced its highest number of recorded homicides. The second hypothesis we stated was that a small number of cities in Brazil were responsible for the recent national increases in homicide. This proved to be the case. Although 2202 of Brazil's 5570 cities experienced increases in homicide between 2016 and 2017, only 30 of these cities accounted for the equivalent national increase in homicides of 5.1%: In Brazil, there was an increase of 3129 homicides between 2016 and 2017; in the 30 cities that experienced the highest increases in homicide between 2016 and 2017, the total increase in these cities was 3131 homicides—equivalent to the national increase; 2172 other cities in Brazil experienced increases in homicide between 2016 and 2017, but these increases were small and offset by the decreases in homicide experienced in many other cities in Brazil.

Studies examining micro-places during periods of increase have suggested that areas where crime concentrates are most responsible for the crime increase. This led to us stating for our third hypothesis that the homicide increase in Brazil was associated with cities that previously recorded the highest levels of homicide. This was not the case for homicides in cities across Brazil—of the cities identified as HHCs, only eight of these were part of the group of 30 cities that accounted for the equivalent national increase in homicides between 2016 and 2017. Instead, several medium-sized and big cities that were not established HHCs in Brazil accounted for the largest proportion of this group of 30 cities. The findings from the current study suggest that although recent increases in homicide in Brazil are highly contained to a relatively small number of cities, the spatial concentration of homicide has dispersed from established areas of high homicide to other cities.

The dispersion of homicide to new problematic cities for homicide does, however, appear to be clustered around several established HHCs. Around Fortaleza, the cities of Caucaia, Horizonte, Maracanaú, Pacajus and Sobral were among the group of 30 cities that accounted for the equivalent national increase in homicides. Around Recife, the cities of Abreu e Lima, Cabo de Santo Agostinho, Gravatá, Ipojuca and Paulista were also cities among this group of 30. Rio de Janeiro and neighboring São Gonçalo were established high homicide cites, and were added to with Duque de Caxias as a city amongst this group of 30 when homicides increased in 2017. Other scholars have suggested there has been a 'reorganization of violence' across Brazil, characterized by the increases in homicide in the north and northeast regions, and from the largest cities to smaller cities [6,53]. The current study supports these patterns of homicide in Brazil, albeit suggesting that several HHCs have continued to persist. As stated in a previous section, Waiselfisz [14,15] proposed five categories to define cities in Brazil where increases in homicide have been observed: new poles of growth, border cities, new frontiers, seaside cities, and cities in the Marijuana Polygon. Based on the results from the current study, we add to this categorization of cities where increases in homicides have been observed in Brazil by suggesting a sixth type of city—*neighboring cities*. This builds on Ingram and Da Costa's [10] observation that homicides in an area are likely to increase homicides in nearby areas.

We define neighboring cities as those that border or are close to established high homicide cities and where conditions are similar for criminal activity to thrive. High levels of homicide in Brazil have dispersed to several cities that border or are close neighbors to high homicide cities. Their proximity to established HHCs is a key influencing factor to why they have emerged as problematic cities for homicide. If these cities were located far from HHCs it is unlikely they would be centers of homicide. By bordering or being close to high homicide cities, the conditions in the neighboring city are more likely to be similar than if the city was located far away. These conditions include the function of the city in terms of commerce, industry and entertainment, social and economic conditions, the effectiveness of government institutions, and the presence of criminal groups. Additionally, these neighboring cities could be where urban expansion from nearby established cities is taking place. Pressures from population movement and limited investment in welfare and public security in these neighboring cities can create environments for criminal activity to thrive [8,14,29,39].

Ten of the 22 cities identified as accounting for the equivalent national increase in homicides between 2016 and 2017 (and which were not HHCs) could be considered as neighboring cities to established HHCs: Caucaia, Horizonte, and Maracanaú because of their proximity to Fortaleza; Abreu e Lima, Cabo de Santo Agostinho, Ipojuca and Paulista because of their proximity to Recife; Ceará-Mirim that borders the HHC of Natal; and Alvorada and Gravataí that border the HHC of Porto Alegre. Additionally, the cluster of cities consisting of Vitória, Cariacica and Serra that are included in the 30 cities group could also be considered as neighboring cities, albeit also being categorized as seaside cities. Pacajus could also be considered a neighboring city, because of it bordering Horizonte and its proximity to the established HHC of Fortaleza, albeit also being categorized as a new pole of growth. All these neighboring cities are likely to be similar in the conditions they experience (to nearby problematic cities for homicide), which in turn offer similar conditions for criminal activity to thrive. Intentional homicide being the ultimate expression of this criminal activity. Table 6 lists cities using the six categories, suggesting that the majority of emerging problematic cities for homicide in Brazil were neighboring cities, with most others being new poles of growth. We also note that almost half of the neighboring cities were also seaside cities, however, this most likely reflects the large geographic distribution in Brazil of urban settlements along or close to the Atlantic Coast.

**Table 6.** Emerging problematic cities that significantly accounted for the equivalent national increase in homicides in Brazil.

| City Categories for Homicide | New Poles of Growth | Border Cities | New Frontiers | Seaside Cities | Marijuana Polygon | Neighboring Cities |
|---|---|---|---|---|---|---|
| **New Poles of Growth** | Barreiras Campinas Gravatá Sobral | | | | | |
| **Border Cities** | | Rio Branco | | | | |
| **New Frontiers** | Altamira | | | | | |
| **Seaside Cities** | | | | Florianópolis Ilhéus | | |
| **Marijuana Polygon** | | | | | | |
| **Neighboring Cities** | Pacajus | | | Cabo de Santo Agostinho Caucaia Ipojuca Paulista Serra Vitória | | Abreu e Lima Alvorada Cariacica Ceará-Mirim Gravataí Horizonte Maracanaú |

The dispersion of crime can also operate in an opposite manner—if high crime areas decreased in crime, this decrease may disperse to neighboring areas. Established high homicide cities such as Belo Horizonte, Brasilia and Curitiba experienced decreases in homicide of between 12% and 20% in 2017. No cities that neighbor these cities experienced increases in homicide that significantly contributed to the national increase, but instead experienced decreases in crime: Contagem located next to Belo Horizonte experienced a 28% decrease in homicides; Formosa located next to Brasilia experienced a 16% decrease in homicides; and São José dos Pinhais, a city that neighbors Curitiba experienced a 12% decrease in homicides.

Policing and public safety programs that are targeted to the micro-places where crime concentration persists have a significant impact in decreasing crime. These programs involve proactive strategies that aim to address the situational causes of crime

(such as deterring criminal activity because of the presence of targeted police patrols), alongside changing individual behaviors that reduce recidivism (e.g., restorative justice), and providing alternatives to criminal involvement (e.g., via focused deterrence strategies). When crime increases, identifying the micro-places most responsible for the increase and targeting activities to these places also has a significant overall impact in decreasing crime [21]. Thus, it is the highly targeted nature of effective intervention implementation that is a key factor in their success. Homicide across cities in Brazil show similar spatial patterns to the patterns of crime observed at micro-places—a small number of places are responsible for a large proportion of homicides, and when homicide increases, a small number of places account for the increase. These patterns provide the opportunity to determine where to target state and national strategies for decreasing homicide, especially during a period of crime increase. Problems of homicide are multi-faceted, but intended program effect can become diluted if not focused on where these programs are most necessary. When these programs are effective, there is the potential for their effect to disperse to neighboring areas. If high homicide levels are not abated, there is the potential for high levels of homicide to disperse to neighboring areas.

Additional analysis was conducted to examine the spatial concentration and dispersion of homicide within city groups. This analysis found that only a small number of cities accounted for a large proportion of homicides in each city group, and that a small number of cities accounted for the equivalent increase in homicides within the city-size group. For example, in the small cities I group, 4% of cities accounted for 25% of homicides, and only 65 of the 3802 cities accounted for the equivalent increase in homicides of 7.5% within this city-size group. The CCDI for small cities I for 2016 to 2017 was 1.5, suggesting that cities other than the high homicide cities within this group were most responsible for the homicide increase. The additional analysis within city-size groups of spatial concentration and the dispersion of homicides during periods of recent homicide increases further showed the redistribution of homicides to the north and northeast regions: 26 of the 28 HHCs in the medium city size category were located in the north or northeast regions, and all those cities that accounted for the equivalent increase in homicides of 6.1% in this city-size group were located in these regions. As other scholars have noted, the increase in violence in the north and northeast regions of Brazil is likely to be associated with changes in the drug trafficking dynamics and the disputes this has created between rival criminal groups, the illegal exploration of land, logging and mining, and land tenure-related conflicts in several areas of these regions [8,12,29,48]. Targeted programs and strategies for effective homicide prevention to the small number of cities in each group would likely result in a more significant decrease in homicides than an untargeted strategy. Although it was beyond the scope of the current study to examine the variables that were most associated with the geographic concentration and dispersion of homicide increase in Brazil (e.g., social inequality and residential stability), it is likely that programs and strategies that are similarly targeted towards addressing the factors that have created the conditions for violence to thrive—such as improving government effectiveness, reducing impunity, and reducing social inequality, and countering the situational circumstances that create opportunities for crime—are likely to be most effective.

## 7. Limitations

There are several limitations that may have affected the study. First, Brazilian homicide figures may be underestimated because of problems associated with misclassification, homicides that are never registered because the body is not found, and structural deficiencies in the criminal justice system that led to the registration of the cause of death as unknown [30,63–65]. According to Cerqueira ([66], p. 42) "it is understood that the homicide rate in the country would be 18.3% higher than the official figures". The number of homicides in Brazil in 2017 was, therefore, more likely to be about 77,000 instead of the 65,602 registered. We do not anticipate this underestimation to have significantly affected the key patterns we observe in the current study. Second, the SIM/MS data on

homicide is made available 18 months after the end of the reporting year. Data for 2017 were released in June 2019, and were the most up to date data available when the current study was conducted.

With regards to methods, using absolute counts drew attention to larger cities. Studies of crime concentration focus on examining where the incidence of crime is greatest, rather than examining crime rates (normalized by population or the size of the geographic unit of study). We referred in the section above to further research we recommend for examining rates of crime in geographic units rather than solely using counts of crime to determine if the patterns observed are different.

The research used cities as the geographic unit of study. The data in some cases referred to an area that extended beyond the physical border of the city. We had no control over this, but as the data have been used in several other studies of cities in Brazil we are confident our results are reflective of the geography of homicides in Brazil at the city level. We also recognize that within cities there is great spatial heterogeneity in social, economic and environmental conditions. This is something that micro-place studies have examined and exposed, and we recognize that the spatial concentration and dispersion of homicide increase is likely to be as similarly acute within cities as the results we show across Brazil in the current study. Crime prevention interventions that target micro-places, such as hot spot policing programs, have been found to significantly decrease crime [17]. A motivation for the current study was to examine if certain cities were more responsible for homicide in Brazil than other cities and examine the spatial dispersion of homicide across Brazil during a period of crime increase. The current study adds to improving our understanding of patterns of homicide in Brazil, and similar to interventions that target micro-places, our results show the potential benefit of targeting strategies and programs to specific cities that contribute most substantially to the problems of homicide within a country.

Since completing the current study we note that homicide levels in Brazil decreased in 2018 and 2019, but increased in 2020. Our study was motivated to examine spatial patterns of homicide during an overall period of homicide increase (from 2007 to 2017). We encourage further research that examines if the findings from our current study and the methods we use provide insight to patterns when homicide decreases and in particular if the decreases were mostly observed in certain cities. In particular, we encourage the use of the typology of cities we add to in the current study (with neighboring cities) and the effect of the Covid-19 pandemic on spatial patterns of homicide in Brazil.

## 8. Conclusions

A small number of cities account for a large proportion of homicides in Brazil. This finding matches with observations of micro-place patterns of crime concentration. When crime increases in micro-places, only a small number of places usually account for the increase in crime, with these places being where crime was previously concentrated. For cities in Brazil, when homicides increased to record levels in 2017, the results from the current study showed that only a small number of cites accounted for the equivalent national increase in crime. However, most of these cities were not established centers of high levels of homicide. Instead, almost all were smaller than the established high homicide cities, albeit many of these new emerging problematic cities for homicide neighbored high homicide cities, especially those located in the northeast of Brazil. Targeted police and public safety programs to the micro-places where crime is observed to concentrate are known to be effective in decreasing crime, especially to counter recent increases in crime. Although the problem of homicides is multi-faceted, there is potential for improving national and state programs and strategies that aim to decrease homicides by more precisely targeting effective interventions to the cities that account for the highest levels of homicide and that are most responsible for a national increase in homicides.

**Author Contributions:** Conceptualization, Spencer P. Chainey; methodology, Spencer P. Chainey and Franklin Epiphanio Gomes de Almeida; software, Spencer P. Chainey and Franklin Epiphanio Gomes de Almeida; validation, Spencer P. Chainey and Franklin Epiphanio Gomes de Almeida; formal

analysis, Spencer P. Chainey and Franklin Epiphanio Gomes de Almeida; investigation, Spencer P. Chainey and Franklin Epiphanio Gomes de Almeida; resources, Spencer P. Chainey and Franklin Epiphanio Gomes de Almeida; data curation, Spencer P. Chainey and Franklin Epiphanio Gomes de Almeida; writing—original draft preparation, Spencer P. Chainey and Franklin Epiphanio Gomes de Almeida; writing—review and editing, Spencer P. Chainey and Franklin Epiphanio Gomes de Almeida; visualization, Spencer P. Chainey; supervision, Spencer P. Chainey; project administration, Spencer P. Chainey and Franklin Epiphanio Gomes de Almeida; funding acquisition, Spencer P. Chainey and Franklin Epiphanio Gomes de Almeida. Both authors have read and agreed to the published version of the manuscript.

**Funding:** Franklin Epiphanio Gomes de Almeida conducted most of his contribution to the research while studying for a MSc in Policing at UCL, and was supported in doing so with funding from a Chevening Scholarship from the UK Foreign, Commonwealth and Development Office. Spencer P. Chainey received no external funding for the research.

**Institutional Review Board Statement:** Not applicable.

**Informed Consent Statement:** Not applicable.

**Data Availability Statement:** Weblinks to the data used are provided in the manuscript.

**Conflicts of Interest:** The authors declare no conflict of interest.

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
