# Peer review of "The Spatial Concentration and Dispersion of Homicide during a Period of Homicide Increase in Brazil"

_ijgi, doi:10.3390/ijgi10080529_

Round 1
Reviewer 1 Report
This study applies the micro-place crime concentration principles to urban geographic units and analyzes the crime pattern and the impact of urban size on crime rate in Brazil. There are some suggestions as the following:
(1) There are some expression problems in lines 28—29, 50-52. Pay attention to word expression and logical relationship.
(2) The legend in Figure 1 is not clear enough, for example, the legend of South is not obvious enough
(3) In Figure 2, the legend setting is not appropriate. The legend of metropolis and big city is larger than that of medium city. If you want to set the legend size according to the city size, metropolis should be larger than big city, and the legend will overlap on the map, so that the location marking is not clear.
(4) It is expected to explain the meaning of ODI and NCDI by explaining their calculation formulas.
(5) There is no further explanation at the end of the article for the three hypotheses (228-231) put forward above. It would be better to echo the beginning and the end.
Author Response
Reviewer 1
R1.1. This study applies the micro-place crime concentration principles to urban geographic units and analyzes the crime pattern and the impact of urban size on crime rate in Brazil. There are some suggestions as the following:
Reply 1.1. We thank you for reviewing the article. Thanks for your comments and advice on how we can further improve the paper. We have made changes you have suggested and think this has resulted in a better article.
R1.2. There are some expression problems in lines 28—29, 50-52. Pay attention to word expression and logical relationship.
Reply 1.2. We have made corrections to the text in the places the reviewer has requested, and in some other places of the text where similar problems were present.
R1.3. The legend in Figure 1 is not clear enough, for example, the legend of South is not obvious enough
Reply 1.3. We have made changes to the legend and presentation of the graph to make it clearer, albeit needing to conform to only using grayscale rather than using colour.
R1.4. In Figure 2, the legend setting is not appropriate. The legend of metropolis and big city is larger than that of medium city. If you want to set the legend size according to the city size, metropolis should be larger than big city, and the legend will overlap on the map, so that the location marking is not clear.
Reply 1.4. We have made a modification to the map by reducing the size of ‘Big cities’ symbol, but as the reviewer recognises we have to be careful with how the symbols overlap each other while at the same time creating symbols that are large enough in size for the information we show on the map to be clear. We believe we have produced a map that does this.
R1.5. It is expected to explain the meaning of ODI and NCDI by explaining their calculation formulas.
Reply 1.5. We have provided a more detailed explanation about how the ODI and NCDI is calculated, with examples showing how this calculation is made. We also further emphasise the original article that describes the Dispersion Calculator, ODI and NCDI so that readers can review this article for further details.
R1.6. There is no further explanation at the end of the article for the three hypotheses (228-231) put forward above. It would be better to echo the beginning and the end.
Reply 1.6. We have now corrected this and felt this was better to do in the discussion section where in the first few paragraphs we reflect on the results and use the three hypotheses we previously stated to do so.
Reviewer 2 Report
This paper, thematically, presents a significant and up-to-date contribution to the literature. It deals with the real issue of the increasing homicide rates in Brazil (and the need to understand its spatial patterns as decision-support to preventive measures), and is in line with the advancements of research in the Criminology of Places and the Geography of Crime.
However, in practice, my major critique is that it feels as though more could have actually been presented, according to the title and the premise of the paper. In fact, most of my concerns (on data, methodological choices and so on) were recognized at the end, in the paper’s limitations, but indeed it would increase the paper’s relevance if some of them could be addressed, if possible, during the analysis.
The case study presentation (including social, political and crime contexts) is very extensive and complete, which contrasts with a very specific theoretical focus, something that at times makes the paper feel repetitive (as certain sentences are repeated in almost every section). This unbalance is also felt later, because the spatial/statistical analysis does not include contextual variables (they are “beyond the scope of the current study”, as written in page 14). So, some of the inferences made in the discussion are not directly connected to the research presented. This also results in that discussions on spatial planning and territorial/social cohesion are left out of the paper, even in the conclusions (the word “planning” is not used once in the paper).
This seems to be an issue that really could improve the reader’s understanding of the paper’s implications. More about territorial organization of Brazil in the early chapters would be useful, especially in comprehending how the country is structured in terms of urban hierarchies (as city size is one of the variables taken into account). One great metropolis would count as “one” city in this study, whereas a polycentric metropolitan area would count as “many” cities of smaller size. Two cities can have the same number of inhabitants and one could be the dormitory periphery of a larger metropolis, and another be the capital of an interior district. Their roles in the urban hierarchy are completely different. Such understanding would have an impact in the way Tables 1 and 2 are read and interpreted. Such questions about city hierarchies and location come immediately to mind when the reader sees in pg. 9: “national increase in homicides was more associated with cities that were not HHCs, rather than HHCs being responsible for the increase”.
At this point, however, the reader does not know that this issue will be taken up in the “Discussion” chapter, even when Fig. 2 is presented. So, when it is somewhat dealt with in pg. 12, it appears almost as an afterthought. It would seem more logical, for the reader, that these questions would be introduced in the “Results”, not the “Discussion” section. It would also enrich the spatial understanding of the problem if a cartographic representations of the indexes calculated could be produced and/or spatial analysis that would take into account location; i.e. put an added focus on spatiality and territorial differences, due to the “national” perspective of the paper.
Precisely on this point, the authors state (pg. 2) that the analysis of spatial patterns of crime concentration have never been applied to “meso geographic units”. I call attention to a recent Portuguese paper (albeit in portuguese) of Saraiva et al (2021), that has presented spatial profiles of 10-year crime statistics, by municipality, for the entire country (https://revistas.rcaap.pt/finisterra/article/view/20682). The two papers take different approaches, but they build on the same theoretical premise of broadening the territorial contexts and spatial-temporal approaches of Crime Concentration to a national planning perspective.
Finally, I just noticed one unintentional mistake. In page 9, Natal says “%” while the other cities say “percent”.
Author Response
Reviewer 2
R2.1. This paper, thematically, presents a significant and up-to-date contribution to the literature. It deals with the real issue of the increasing homicide rates in Brazil (and the need to understand its spatial patterns as decision-support to preventive measures), and is in line with the advancements of research in the Criminology of Places and the Geography of Crime.
Reply 2.1. We thank you for reviewing the article. Thanks for your comments and advice on how we can further improve the paper. We have made changes you have suggested and think this has resulted in a better article.
R2.2. However, in practice, my major critique is that it feels as though more could have actually been presented, according to the title and the premise of the paper. In fact, most of my concerns (on data, methodological choices and so on) were recognized at the end, in the paper’s limitations, but indeed it would increase the paper’s relevance if some of them could be addressed, if possible, during the analysis.
The case study presentation (including social, political and crime contexts) is very extensive and complete, which contrasts with a very specific theoretical focus, something that at times makes the paper feel repetitive (as certain sentences are repeated in almost every section). This unbalance is also felt later, because the spatial/statistical analysis does not include contextual variables (they are “beyond the scope of the current study”, as written in page 14). So, some of the inferences made in the discussion are not directly connected to the research presented. This also results in that discussions on spatial planning and territorial/social cohesion are left out of the paper, even in the conclusions (the word “planning” is not used once in the paper).
Reply 2.2. The research we present in this article was substantial, albeit with a focus towards the geographic dynamics of homicides at the city level across Brazil. To date, most studies of micro-place crime concentration only examines this pattern, and most studies relating to the increases and emerging problem places only examine these particular patterns. In the current study we do both and think that by doing so our study makes a valuable contribution to the literature, and which we believe is suitable for the International Journal of Geographic Information.
However, we have made revisions to the revised manuscript that provides more of a focus to the geographic dynamics (i.e., spatial concentration and the spatial dispersion of crime increase) of homicide in Brazil, and removed some of the text relating to the determinants of homicide. On review, we felt the description in the original manuscript in section 2 about ‘Homicides in Brazil and its Determinants’ distracted the reader from what was the main aim and focus of our study. By doing so, this better places the content of the discussion, limitations and further research sections in which we explain that next steps could involve examining our findings in relation to factors such as social inequality, social disorganization and urbanization.
R2.3. This seems to be an issue that really could improve the reader’s understanding of the paper’s implications. More about territorial organization of Brazil in the early chapters would be useful, especially in comprehending how the country is structured in terms of urban hierarchies (as city size is one of the variables taken into account). One great metropolis would count as “one” city in this study, whereas a polycentric metropolitan area would count as “many” cities of smaller size. Two cities can have the same number of inhabitants and one could be the dormitory periphery of a larger metropolis, and another be the capital of an interior district. Their roles in the urban hierarchy are completely different. Such understanding would have an impact in the way Tables 1 and 2 are read and interpreted. Such questions about city hierarchies and location come immediately to mind when the reader sees in pg. 9: “national increase in homicides was more associated with cities that were not HHCs, rather than HHCs being responsible for the increase”.
At this point, however, the reader does not know that this issue will be taken up in the “Discussion” chapter, even when Fig. 2 is presented. So, when it is somewhat dealt with in pg. 12, it appears almost as an afterthought. It would seem more logical, for the reader, that these questions would be introduced in the “Results”, not the “Discussion” section. It would also enrich the spatial understanding of the problem if a cartographic representations of the indexes calculated could be produced and/or spatial analysis that would take into account location; i.e. put an added focus on spatiality and territorial differences, due to the “national” perspective of the paper.
Reply 2.3. We felt that is was not our place in the current study to determine urban hierarchies, but instead draw from the concepts and definitions from previous similar studies to posit our study. That is, use the definitions that have been created in Brazil for metropolis and other cities (according to their size – as proposed by Duarte et al. (2012) and that we use and quote in the current study) and use the definitions/five categories that Waiselfisz (2011; 2015) proposed. We think the results we present in the current study are sufficiently substantial (as refered to in Reply 2.2) for publication in a single article.
However, we do recognise that the discussion in which we use and then add to Waiselfisz five categories was not set up that well in the original manuscript. To address this we have added text in the paragraph before the Data and Methods section (copied below) so that it is clearer that the important points we make in the discussion about new poles of growth, new frontiers etc, and our introduction of ‘neighboring cities’ is not interpreted as an ‘afterthought’:
“The research was guided by testing three hypotheses: homicide is highly concentrated across cities in Brazil; a small number of cities in Brazil were responsible for recent national increases in homicide; the homicide increase in Brazil was associated with cities that previously recorded the highest levels of homicide. We anticipate that the results that are generated from testing these hypotheses provide new insights into the geographic dynamics of homicide in Brazil. To posit these results in the context of previous research that has examined the patterning of homicides in Brazil we aim to review our results in the discussion section against Waiselfisz (2011) five categories for defining cities in Brazil where increases in homicide have been observed.”
With regards to the “cartographic representations of the indexes”, there is no simple way to do this, particularly with a dataset consisting of over 5000 geographic units. The indices we use in the research are designed to generate the statistical output that we include in the paper (and shown in table 3) and to highlight those geographic units (i.e. cities) that contributed most to an increase in crime. Figure 2 does the latter, using the results of the indices to show those cities that accounted for the equivalent national increase in homicides between 2016 and 2017.
R2.4. Precisely on this point, the authors state (pg. 2) that the analysis of spatial patterns of crime concentration have never been applied to “meso geographic units”. I call attention to a recent Portuguese paper (albeit in portuguese) of Saraiva et al (2021), that has presented spatial profiles of 10-year crime statistics, by municipality, for the entire country (https://revistas.rcaap.pt/finisterra/article/view/20682). The two papers take different approaches, but they build on the same theoretical premise of broadening the territorial contexts and spatial-temporal approaches of Crime Concentration to a national planning perspective.
Reply 2.4. We were not aware of this very recent publication because our review of the literature was completed before this article was published. We thank the reviewer for this information. We have read the paper and have now cited it in the revised manuscript.
R2.5. Finally, I just noticed one unintentional mistake. In page 9, Natal says “%” while the other cities say “percent”.
Reply 2.5. Corrected.
Reviewer 3 Report
The issue of homicides is a very serious global problem that can disorganize an entire society. In fact, in some parts of the world, including Brazil, it has become a scourge. Well, the honorable authors with their interesting article put a priority on the big problem of this country, trying to give in addition to its sociological, political, racial, and cultural interpretation a geographical one respectively, which is extremely important and works to combat it.
In general, the article has a sound scientific structure and a well-documented methodology, with references which for the most part are from the last eight years. The wording of the research is sound and thorough without unnecessary references and the discussion section shows that the conclusions that have emerged from the processing of the data are well documented and are aimed precisely at the problem discussed here.
Nevertheless, I identified some - mainly methodological - issues that I do not consider giving the article the required scientific soundness. The authors take into account the studies that have been done from time to time for different regions of Brazil but on a small or very small scale whose methodological rules are utilized here on a city scale, which is the spatial unit of the present research. At the same time, while trying to explain descriptively the indicators produced by the above approach, they do not proceed to a formulation of the statistical methods and algorithms they use, so that in the end it can be judged whether it is possible to use them in different geographical scales. In my opinion, this is a serious methodological omission.
The simple map they have constructed (p. 11) to mark the areas that show the increase in homicides shows a simplified classification of cities into three categories and, of course, their location. So here, in my opinion, there should be an additional quantitative distinction of the magnitude of the increase in homicides (eg variable radius cycles), so that the reader understands the severity of the problem by city. In addition, what is clear is that the problem of rising homicides occurs mostly on the east side of the country and even near or on its long coastline, something that has received a little comment, although geographically it is a burning issue. On the contrary, from the beginning of the development of their methodology, the esteemed authors seem to be geographically biased towards the areas they examined, classifying them as north, northeast, southeast, south, and mid-west. I consider that at a statistical level at least the above ranking is not correct as it seems that the null hypothesis is ignored.
Another observation is that there is no reference to the cause of the homicides under investigation or any classification. It is very important, for example, to know whether most crimes are due to drug trafficking, and so on. those that are due to trafficking or any other reason, as the techniques of dealing with and/or preventing one or the other differ.
Two additional small remarks are that the unit of analysis may not be the total of 5570 cities, but one city (line 233), as well as that the word "currency" should be replaced with the word "circulation" ( line 253).
In conclusion, I believe that this article is important, and its publication is very important, as long as the necessary corrections are made.
Best regards
Author Response
Reviewer 3
R3.1. The issue of homicides is a very serious global problem that can disorganize an entire society. In fact, in some parts of the world, including Brazil, it has become a scourge. Well, the honorable authors with their interesting article put a priority on the big problem of this country, trying to give in addition to its sociological, political, racial, and cultural interpretation a geographical one respectively, which is extremely important and works to combat it.
In general, the article has a sound scientific structure and a well-documented methodology, with references which for the most part are from the last eight years. The wording of the research is sound and thorough without unnecessary references and the discussion section shows that the conclusions that have emerged from the processing of the data are well documented and are aimed precisely at the problem discussed here.
Reply 3.1. We thank you for reviewing the article. Thanks for your comments and advice on how we can further improve the paper. We have made changes you have suggested and think this has resulted in a better article.
R3.2. Nevertheless, I identified some - mainly methodological - issues that I do not consider giving the article the required scientific soundness. The authors take into account the studies that have been done from time to time for different regions of Brazil but on a small or very small scale whose methodological rules are utilized here on a city scale, which is the spatial unit of the present research. At the same time, while trying to explain descriptively the indicators produced by the above approach, they do not proceed to a formulation of the statistical methods and algorithms they use, so that in the end it can be judged whether it is possible to use them in different geographical scales. In my opinion, this is a serious methodological omission.
R3.2. We have provided a more detailed description and explanation about how the crime concentration measures were calculated and how the ODI and NCDI were calculated, with examples showing how these calculations were made. We also further emphasise the original article that describes the Dispersion Calculator, ODI and NCDI so that readers can review this article for further details and to assist them in using the Dispersion Calculator software to replicate the processes we used.
R3.3. The simple map they have constructed (p. 11) to mark the areas that show the increase in homicides shows a simplified classification of cities into three categories and, of course, their location. So here, in my opinion, there should be an additional quantitative distinction of the magnitude of the increase in homicides (eg variable radius cycles), so that the reader understands the severity of the problem by city. In addition, what is clear is that the problem of rising homicides occurs mostly on the east side of the country and even near or on its long coastline, something that has received a little comment, although geographically it is a burning issue. On the contrary, from the beginning of the development of their methodology, the esteemed authors seem to be geographically biased towards the areas they examined, classifying them as north, northeast, southeast, south, and mid-west. I consider that at a statistical level at least the above ranking is not correct as it seems that the null hypothesis is ignored.
Reply 3.3. We gave much thought to how to cartographically present the key results and based on other comments we decided not to use graduated symbols. Each city, as described in the figure caption and referred to in text of the manuscript is one of the 30 cities in Brazil (out of over 5,000) that accounted for the equivalent national increase in homicides between 2016 and 2017. So, as they are all is this same category we thought (and tested) that showing each as a graduated symbol added little because they all experienced levels of homicide that were similar in most cases. Also, when we tested the use of graduated symbols to represent the number of homicides it meant that larger symbols overlapped smaller symbols and made it difficult to see all cities that are included on the map.
R3.4. Another observation is that there is no reference to the cause of the homicides under investigation or any classification. It is very important, for example, to know whether most crimes are due to drug trafficking, and so on. those that are due to trafficking or any other reason, as the techniques of dealing with and/or preventing one or the other differ.
Reply 3.4. We have corrected this by stating the following in the revised manuscript:
“The number of homicides in each city between 2007 and 2017 was extracted from the Mortality Information System of the Brazilian Ministry of Health (SIM/MS, 2019). These data refer to the occurrence of violent deaths, including intentional incidents, robberies that resulted in a homicide, and police killings. The data do not indicate the motivation for the homicide, such as if the intentional homicide was associated with the drugs trade and violence between criminal groups.”
R3.5. Two additional small remarks are that the unit of analysis may not be the total of 5570 cities, but one city (line 233)
Reply 3.5. Corrected
R3.6. as well as that the word "currency" should be replaced with the word "circulation" ( line 253).
Reply 3.6. Rather than ‘circulation’ we have changed ‘currency’ to ‘recency’.
R3.7. In conclusion, I believe that this article is important, and its publication is very important, as long as the necessary corrections are made.
Reply 3.7. Again, we thank the reviewer and hope our revisions have met with their suggested changes, or at least our replies have answered questions they made.
Round 2
Reviewer 3 Report
In the new improved version of the article, the honorable authors do not think that they have answered all the questions that were asked to them in a clear and unambiguous way.
More specifically the question "... At the same time, while trying to explain descriptively the indicators produced by the above approach, they do not proceed to a formulation of the statistical methods and algorithms they use, so that in the end it can be judged whether it is possible to use them in different geographical scales ... "I consider that it has not been fully understood and perhaps it was not answered as one would expect. At this point, I was not asking to simply mention Dispersion Calculator, but to explain how extrapolation was done from neighborhood-scale data to city-data. This is statistically feasible, but it still needs to be explained. Also, the Dispersion Calculator is treated as a "closed box" without showing the algorithms it uses so that it can be evaluated that its results are indeed reliable.
Another point which I had touched and did not answer was: "... In addition, what is clear is that the problem of rising homicides occurs mostly on the east side of the country and even near or on its long coastline, something that has received a little comment, although geographically it is a burning issue. On the contrary, from the beginning of the development of their methodology, the esteemed authors seem to be geographically biased towards the areas they examined, classifying them as north, northeast, southeast, south, and mid-west. I consider that at a statistical level at least the above ranking is not correct as it seems that the null hypothesis is ignored "...
Finally, I still believe that the map that appears can and should be reconstructed in such a way as to quantify the magnitude of homicides along with the classification of cities into three categories.
Author Response
Reviewer 3
R3.2.1. In the new improved version of the article, the honorable authors do not think that they have answered all the questions that were asked to them in a clear and unambiguous way.
Reply 3.2.1. We thank the reviewer for their additional review. We believe we have now addressed all the points they have raised, with details provided below.
R.3.2.2. More specifically the question "... At the same time, while trying to explain descriptively the indicators produced by the above approach, they do not proceed to a formulation of the statistical methods and algorithms they use, so that in the end it can be judged whether it is possible to use them in different geographical scales ... "I consider that it has not been fully understood and perhaps it was not answered as one would expect. At this point, I was not asking to simply mention Dispersion Calculator, but to explain how extrapolation was done from neighborhood-scale data to city-data. This is statistically feasible, but it still needs to be explained. Also, the Dispersion Calculator is treated as a "closed box" without showing the algorithms it uses so that it can be evaluated that its results are indeed reliable.
Reply 3.2.2. There are a number of parts to our response to this comment. First, a point of clarification and then we provide the full details, from the revised manuscript, about the description of how the Dispersion Calculator and the calculation of the ODI and NCDI.
First, to clarify, there is no ‘extrapolation’ from neighborhood-scale data to city-data in how the Dispersion Calculator works. We explain this as follows (on page 8 in the revised manuscript) and in particular with regards to our statement that ‘their application is suitable for any size of geographic unit, such as cities’. Please also see the additional comment about this at the end of reply 3.2.2:
“To help measure and identify the areas that are most responsible for an overall increase in crime, Ratcliffe (2010) developed a series of dispersion indices to indicate if an overall crime increase is associated with only a small number of areas or if the increase is a spreading (emergent) problem. Ratcliffe’s dispersion measures do not, however, determine if the crime increase is associated with an increase in areas where most crime previously occurred. To address this, Chainey and Monteiro (2019) developed the Crime Concentration Dispersion Index (CCDI) to determine whether, during a period of crime increase, areas of high crime concentration were responsible for the increase or if other areas were responsible. To date, these measures for crime dispersion have only been applied to micro-place and neighborhood geographic units (because of recent research focus to micro-place analysis of crime), yet their application is suitable for any size of geographic unit, such as cities. The techniques developed by Ratcliffe (2010) and Chainey and Monteiro (2019), therefore, can be used to determine whether an increase in crime in a country was mainly associated with only a small number of cities being responsible for the increase, and whether cities with the highest levels of crime were mainly responsible for a national increase in crime.
In the revised manuscript, in the Data and Methods section, we next describe how the Dispersion Calculator works (i.e.,. it’s algorithm) as follows:
“The study’s examination of the dispersion of homicide was focused towards examining the change in crime between the two years for the most recent period that data were available (i.e., 2016 to 2017) for purposes of recency, and because 2017 was when there was a peak in the number of homicides observed in Brazil. The spatial dispersion of homicide increase in Brazil was analysed using Ratcliffe's Dispersion Calculator (2010). The Dispersion Calculator compares the changes in crime between two time periods (t1 and t2 i.e., 2016 and 2017) for each geographic unit in the study area. When a crime increase in a study area has been observed, the Dispersion Calculator determines whether the increase in the study area was related to only a small number of places experiencing a large increase in crime, or whether the increase was associated with smaller increases across a large number of places within the study area. The Dispersion Calculator works by ordering the geographic units in the study area by the level of crime increase in each unit, and then removes these ordered units one at a time (starting with the unit with the highest crime increase) and removes their respective frequency of crime from the total for the study area. For each iteration, the increase in crime across the study area (based on all remaining geographic unit) is recalculated. By doing so, the Dispersion Calculator determines the point at which the removal of the geographic units that experienced the highest increases in crime generates a revised study area change in crime measure that shows no increase in crime across the remaining geographic units between the two time periods. Some other geographic units may have experienced an increase in crime, but these increases would have been smaller and could have been offset by the places where decreases in crime were observed.”
We also ask the reviewer to recall that the Dispersion Calculator is not our invention. We believe we provide sufficient information in our manuscript to allow researchers to replicate our methods whilst also reading the original article that describes the Dispersion Calculator and the Dispersion Calculator software.
We then describe the ODI as follows and illustrate with an example:
“The Dispersion Calculator generates two indices: the Offence Dispersion Index (ODI) and the Non-Contributory Dispersion Index (NCDI). The ODI is the proportion of geographic units that must be removed from the study-wide calculation before the increase in crime across the study area is transformed to a decrease, or at least a no-change steady state is observed (for more details see Ratcliffe, 2010). The ODI is calculated using the following equation:
ODI = (n of geographic units that must be removed from the study-wide calculation before the increase across the study area is transformed to a decrease)/(n of all geographic units in the study area)
The ODI (ranging from zero to one) determines the smallest proportion of geographic units that alone account for a study area equivalent increase in crime. For example, in a study area that experienced a 20 percent increase in crime and consisted of 100 geographic units, the ODI is a measure of the smallest proportion of geographic units that alone accounted for the study area’s 20 percent increase in crime. For instance, if five geographic units experienced large increases in crime, and the total increase in these five geographic units was equivalent to the study area’s 20 percent increase in crime, the ODI would be 0.05 (i.e., 5/100). If ten geographic units experienced the largest increases that in total was equivalent to the study area’s 20 percent increase in crime, the ODI would be 0.1 (i.e., 10/100). An ODI value close to zero indicates that only a small number of geographic units experienced an increase in crime that was equivalent to the study area’s increase.”
We then describe the NCDI as follows and illustrate with an example:
“The NCDI (ranging from zero to one) indicates the proportion of other geographic units that are a concern and is a measure of the proportion of the other geographic units in the study area that experienced an increase in crime. The NDI is calculated using the following equation:
NCDI = (n of geographic units in the study area that experienced an increase in crime, but not including those units included in the ODI calculation that experienced the highest increases in crime)/(n of all geographic units in the study area)
For example, following on from the example of a study area that consists of 100 geographic units, if 35 other geographic units experienced increases in the crime (i.e., in addition to the five geographic units that experienced the greatest increases in crime), the NCDI for this study area would be 0.35 (i.e., 35/100). An NCDI value close to zero indicates the crime increase has not been observed in many other geographic units.”
Next, we provide further clarification that we hope additionally addresses the reviewer’s comment “so that in the end it can be judged whether it is possible to use them in different geographical scales” by stating the Dispersion Calculator can be applied to any geographic unit of any type
“To date, although the ODI and NCDI can be applied to any size of geographic unit and any size of study area it has only been applied to the geographic units of street segments and neighbourhoods (for examples see Chainey and Monteiro, 2019; Ratcliffe, 2010). In the current study we apply these measures to the geographic unit of cities.”
We hope our comments on points of clarification and details about the revisions we have made in the third version of the manuscript are acceptable to the reviewer.
R.3.2.3. Another point which I had touched and did not answer was: "... In addition, what is clear is that the problem of rising homicides occurs mostly on the east side of the country and even near or on its long coastline, something that has received a little comment, although geographically it is a burning issue. On the contrary, from the beginning of the development of their methodology, the esteemed authors seem to be geographically biased towards the areas they examined, classifying them as north, northeast, southeast, south, and mid-west. I consider that at a statistical level at least the above ranking is not correct as it seems that the null hypothesis is ignored "...
Reply 3.2.3. We recognise the reviewers points, and first provide some comments that relate to the focus of the study before providing comments and information about revisions that relate to the regional descriptions we use.
The focus of the study was based around testing three hypotheses:
“homicide is highly concentrated across cities in Brazil; a small number of cities in Brazil were responsible for recent national increases in homicide; the homicide increase in Brazil was associated with cities that previously recorded the highest levels of homicide”.
These hypotheses reflect the motivation for the study, and hence our focus by organizing the analysis in the way we did. As explained in the manuscript, we also examined the results by grouping cities in Brazil by city size, as follows:
“Cities were organized into five size groupings to determine if patterns of homicide were particular to city size, following the method used by Duarte et al. (2012): …!
We then ask the reviewer to recall that we provide several examples in the manuscript where we use terms to describe regions in Brazil that are the terms that other researchers have previously used:
“Coinciding with the role of Brazil as a hub for drug trafficking has been the geo-economic expansion of the largest criminal organizations in the country - the Primeiro Comando da Capital (PCC) and the Comando Vermelho (CV). These groups, previously most present in the southeast and south regions of Brazil, have expanded their activities to the north and northeast of the country with allies based in these regions (Garzón-Vergara, 2016; IPEA, 2019a; Manso and Dias, 2018). Collectively, these studies have provided valuable insights into the variation and dynamics of homicide in Brazil.”
“Other scholars have suggested there has been a ‘reorganization of violence’ across Brazil, characterized by the increases in homicide in the north and northeast regions, and from the largest cities to smaller cities (Andrade & Diniz, 2013; Steeves et al., 2015).”
We then state at the end of the Data and Methods section:
“To assist in the presentation of the results we also organised the results for cities by the regional geography of Brazil (north, northeast, southeast, south and midwest). We used these regions in order to be consistent with the definition of regions of Brazil that are most commonly used, particularly in studies of homicide in Brazil (IPEA 2019).”
So, primarily for purposes of consistency with previous research on homicides in Brazil we use the regional definitions that other researchers have used so that our research complements/builds on this other research. The latter is a point that other reviewers have raised to us, stating in particular that our choice of regions (in how we organised the presentation of results) should conform with one of the major annual studies on homicides in Brazil - the annual Atlas of Violence of Brazil (IPEA 2019; 2018) – which organises the presentation of data on homicides into the same regional groupings we use. If we had referred to the ‘east side of the country’ this would have been at odds with the common regional definitions of Brazil and would have caused confusion.
However, as the reviewer observes and which is a point that we also make at the end of the results section, many of the HCCs are located along or near to the Atlantic Coast. However, this does primarily reflect the geographic distribution of urban settlements in Brazil, and we have included the following in the revised manuscript to acknowledge this:
“Over half of these 30 cities were located in the northeast region of Brazil, and almost all the cities were located along the Atlantic coast. However, it is noted that the majority of urban settlements in Brazil are located along or near to the Atlantic coast, reflecting the lack of properly developed transport routes that extend into the interior from what is referred to as the Grand Escarpment that dominates much of Brazil’s coast (Marshall, 2015).”
We then take our observations about the geographic distribution of the HCCs we identify by following definition of cities that were defined Waiselfisz, and which includes discussion of cities along the coast. Waiselfisz describes these cities as ‘seaside cities’. We also use this definition but we do so in the context of other city definitions that Waiselfisz uses and in the context of recognising the overall geographic distribution of urban settlements in Brazil. This is how we base or discussion of the results, but do so by also stating (with reference to Table 6):
“We also note that almost half of the neighboring cities were also seaside cities, however, this most likely reflects the large geographic distribution in Brazil of urban settlements along or close to the Atlantic Coast.
We believe we offer a more valuable contribution to the research literature on homicide in Brazil by using Waiselfisz definitions of cities to discuss our results, which we then add to by defining ‘neighboring cities’ as a new category.
R.3.2.4. Finally, I still believe that the map that appears can and should be reconstructed in such a way as to quantify the magnitude of homicides along with the classification of cities into three categories
Reply 3.2.4. We have created a new map to also show homicide increase for each category of city size. Please bear in mind we also had to conform with the request of another review who did not ask us to include graduated symbols to represent the increase in homicides but did require us to create symbols that respected the different sizes of the cities. We have, therefore, done both in the revised map we have created, without creating the issue of large graduated symbols overlapping/obscuring small symbols (for smaller cities where increases in homicides were lower).